# The Importance of Skills Training for Service Users in Health Care: A Comparative Analysis

**DOI:** 10.3390/healthcare10081554

**Published:** 2022-08-16

**Authors:** Abraham Rudnick

**Affiliations:** Departments of Psychiatry and Bioethics and School of Occupational Therapy, Dalhousie University, Dartmouth, NS B3B 1Y6, Canada; harudnick@hotmail.com

**Keywords:** education, health care, service users, skills training

## Abstract

Health care and other human services such as education are diverse in many substantive aspects. However, arguably, they all use the skills training of service users to accomplish at least some of their key aims. This brief article uses a selective literature review and a comparative analysis to review the commonalities of skills training of service users in health care and education in order to illustrate and argue for the importance of a skills training approach in health care and other human services (recognizing that complementary approaches such as support provision are needed too and that not all service users need skills training). Research and development is needed in this area.

## 1. Introduction

Human services contribute to people surviving and thriving. For example, health care services, such as prevention, treatment and rehabilitation, advance and sustain people’s health and wellbeing, and education services such as elementary education and professional training advance and sustain people’s satisfaction and success. These services clearly differ substantively from each other, yet as noted above, they are similar in some of their ends. Are they also similar in some of their means? This question has not been addressed much as it requires a trans-disciplinary approach, yet answering it may inform and improve such human services, e.g., by further cross-fertilization across them. This brief article uses a selective literature review and comparative analysis to review the commonalities of skills training of service users across health care and other-education-human services in order to illustrate and argue for a skills training approach to them (recognizing that not all service users may need such skills training and that complementary approaches are needed too, such as a supports provision approach; supports—unlike skills— are external to a person, i.e., they are performed by the person’s environment, e.g., workplace accommodations [1], and can be synergistic with skills [2]).

## 2. A Skills Training Approach

As noted above, skills are internal to a person. That is, they are actions performed by the person rather than by their environment. Admittedly, persons and environments are interactive, e.g., they are influenced by each other, so a person’s skills can be influenced by their environment and vice versa [2]. Skills are abilities to act. The three most general categories of skills are behavioral, cognitive and emotional skills, with different areas of research focusing on different sets of skills within this classification [3,4]. Examples of behavioral skills are gross motor skills such as athletic skills and fine motor skills such as surgical skills. Examples of cognitive skills are intellectual skills such as critical thinking skills and perceptual skills such as wine tasting skills. Examples of emotional skills are intra-personal skills such as self-calming skills and inter-personal skills such as empathy skills (which are combined with behavioral—expressive—skills to form communication skills). A skills training approach involves focusing primarily, if not exclusively, on training people to acquire and use skills as effectively (including, among other things, safely) and efficiently (including, among other things, rapidly) as possible.

## 3. Skills Training of Service Users in Health Care

Health care involves both skills training and supports provision to service users in order to be person-centered as well as evidence-based [5]. In relation to prevention, skills training involves training service users to eat healthy and to exercise physically, among other things, and supports provision involves vaccinations, among other things. In relation to treatment, skills training involves training service users to take their medications as prescribed and to communicate effectively with their health care providers, and supports provision involves medications, among other things. In relation to rehabilitation, skills training involves training service users to enhance their compensatory strengths and to normalize their lives as much as possible, and supports provision involves assistive devices, among other things. Skills training is required in various circumstances for health improvement, e.g., information provision, which is a type of support provision, has to be combined with skills training in adaptive (behavioral, cognitive and emotional) coping across prevention, treatment and rehabilitation for sustained positive outcomes [6].

## 4. Skills Training of Learners in Education Services

Education services for various ages and at various stages of learners traditionally focus on academic and professional skills training, and supports provision, such as in mathematics, sciences, technology, humanities, arts, and trades. Rarely (if ever, in some jurisdictions) have education services addressed other skills training, as much as it is needed. For example, many children need intra-personal and inter-personal skills training such as emotional and social skills training to function well (not just academically) [7]. In addition to skills training for children in education services, adults involved with children often need related skills training. For example, some parents need communication skills training to parent their children well [8].

Arguably, many people need various social and other non-professional skills training services to succeed and to be satisfied beyond their professional work. This seems self-evident in relation to people’s health care environments, as many people with health challenges need to learn and use critical thinking skills and assertive communication skills in order to engage effectively with their health care providers and the health care system more generally. For example, research has shown that such skills training for people with mental health challenges improves their health outcomes; unfortunately, this skills training focuses primarily on their mental health services, and less so or not at all on their other health services such as their physical health services [9,10]. Thus, health care can be viewed as a set of learning environments for service users, particularly in relation to health care that addresses chronic health challenges that require active engagement of service users for their optimal effectiveness.

## 5. Conclusions

Health and education services, and likely other human services such as social services, require a range of behavioral, cognitive and emotional skills training that allows their service users to perform well and to be well, recognizing that not all service users may need such skills training. This brief opinion paper is limited as it is suggestive rather than conclusive, considering its selective rather than systematic literature review and its illustrative rather than comprehensive comparative analysis. Research and development is much needed in this area, including for improved health care.

## Data Availability

Not applicable.

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
