# Peer review of "The Importance of Skills Training for Service Users in Health Care: A Comparative Analysis"

_healthcare, 2022, doi:10.3390/healthcare10081554_

Round 1
Reviewer 1 Report
The manuscript aims to identify some basic skills for the training of staff in different fields.
However, the level of theoretical background is very weak, and there is neither a methodological basis for the study nor a presentation of results that would allow the contributions to be seen in an illustrative way.
In conclusion, the work does not represent a relevant or academically grounded advance in the area of knowledge.
Author Response
Thank you for your review. This was submitted as an opinion paper so your commentary re research design and more generally methodology is not applicable. As you do not provide substantive suggestions for revision I cannot substantively revise the paper in relation to your review. I have edited the paper linguistically.
Reviewer 2 Report
Thank you for the opportunity to review this opinion paper. I think it is well written, but I would argue that your conclusion is to far fetched.
Although some if not most service users may benefit from some skills training not every service user will need an emotional skill training to do or be well. I suggest you to clarify your conclusion on that.
Good luck.
Author Response
Thank you for your review. I have revised my conclusion accordingly to state that not every service user may need such skills training. I have edited the paper linguistically.
Reviewer 3 Report
Skills training for healthcare users is undoubtedly a topic of interest.
This paper is just an opinion piece, well written. In my opinion it cannot be submitted to peer review since it is not a scientific article, it would be preferable for the editors to make the decision to publish it or not.
Additional comments: The paper contains numerous statements that are not supported by bibliographic sources. I suggest that, despite the brevity of the article, each of the comments made by the author be reinforced with a scientific basis.
Author Response
Thank you for your review. I have added references accordingly, within the constraints of the standard of a small number of references for such opinion papers. I have also edited the paper linguistically.
Round 2
Reviewer 3 Report
The paper has been improved and new bibliographic references have been incorporated that give it greater rigor.